# A Fatty Diet Induces a Jejunal Ketogenesis Which Inhibits Local SGLT1-Based Glucose Transport via an Acetylation Mechanism—Results from a Randomized Cross-Over Study between Iso-Caloric High-Fat versus High-Carbohydrate Diets in Healthy Volunteers

**DOI:** 10.3390/nu14091961

**Published:** 2022-05-07

**Authors:** Erik Elebring, Ville Wallenius, Anna Casselbrant, Neil G. Docherty, Carel W. le Roux, Hanns-Ulrich Marschall, Lars Fändriks

**Affiliations:** 1Institute of Clinical Sciences, Department of Surgery, Sahlgrenska Academy, University of Gothenburg, SE41345 Gothenburg, Sweden; erik.elebring@gu.se (E.E.); ville.wallenius@gastro.gu.se (V.W.); anna.casselbrant@gastro.gu.se (A.C.); 2Department of Surgery, Sahlgrenska University Hospital, SE41345 Gothenburg, Sweden; 3Metabolic Medicine, School of Medicine, Conway Institute, University College Dublin, D04 V1W8 Dublin, Ireland; neil.docherty@ucd.ie (N.G.D.); carel.leroux@ucd.ie (C.W.l.R.); 4Institute of Medicine, Department of Molecular and Clinical Medicine, Sahlgrenska Academy, University of Gothenburg, SE41345 Gothenburg, Sweden; hanns-ulrich.marschall@gu.se; 5Department of Medicine, Sahlgrenska University Hospital, SE41345 Gothenburg, Sweden

**Keywords:** jejunum, mucosa, monosaccharides, histone acetylation

## Abstract

Background and aims: Insights into the nature of gut adaptation after different diets enhance the understanding of how food modifications can be used to treat type 2 diabetes and obesity. The aim was to understand how diets, enriched in fat or carbohydrates, affect glucose absorption in the human healthy jejunum, and what mechanisms are involved. Methods: Fifteen healthy subjects received, in randomised order and a crossover study design, two weeks of iso-caloric high-fat diet (HFD) and high-carbohydrate diet (HCD). Following each dietary period, jejunal mucosa samples were retrieved and assessed for protein expression using immunofluorescence and western blotting. Functional characterisation of epithelial glucose transport was assessed ex vivo using Ussing chambers. Regulation of SGLT1 through histone acetylation was studied in vitro in Caco-2 and human jejunal enteroid monolayer cultures. Results: HFD, compared to HCD, decreased jejunal Ussing chamber epithelial glucose transport and the expression of apical transporters for glucose (SGLT1) and fructose (GLUT5), while expression of the basolateral glucose transporter GLUT2 was increased. HFD also increased protein expression of the ketogenesis rate-limiting enzyme mitochondrial 3-hydroxy-3-methylglutaryl-CoA synthase (HMGCS2) and decreased the acetylation of histone 3 at lysine 9 (H3K9ac). Studies in Caco-2 and human jejunal enteroid monolayer cultures indicated a ketogenesis-induced activation of sirtuins, in turn decreasing SGLT1 expression. Conclusion: Jejunal glucose absorption is decreased by a fat-enriched diet, via a ketogenesis-induced alteration of histone acetylation responsible for the silencing of SGLT1 transcription. The work relates to a secondary outcome in ClinicalTrials.gov (NCT02088853).

## 1. Introduction

The intestinal absorption of monosaccharides is one of the fundamental mechanisms in the entry of new energy into the body. Still, we do not know in detail the mechanisms that regulate the uptake of the prominent monosaccharide glucose via the intestinal mucosa. It is well known that enzymes in the intestinal fluid degrade ingested complex dietary carbohydrates into mono- or disaccharides. These in turn are absorbed by mono- or disaccharide transporters attached to the intestinal mucosa. Among these, particularly the sodium-glucose linked transporter 1 (SGLT1) transports glucose and galactose into the enterocyte in co-transport with sodium. In addition to being a monosaccharide transporter, the SGLT1 is also known as a water transporter, probably due to its sodium transport [1]. Other monosaccharide transporters, for example, the basal epithelial glucose transporter 2 (GLUT2), then allow the intracellular glucose to leave the basolateral membrane and eventually enter the bloodstream [2]. Once glucose has been absorbed by the intestinal epithelium, it will be either immediately metabolized to energy, stored in the organism as more complex carbohydrates (for example glycogen), or converted into fat for later disposal. An increased presence of glucose in the intestinal lumen increases the SGLT1 expression in the luminal aspect of the enterocytes and *vice versa*. However, it remains controversial which are the mechanisms that link intestinal glucose availability to the local SGLT1 expression in the intestinal mucosa [3]. Due to its inaccessibility, the normal gut mucosa has not been extensively studied in humans [4]. Data in the literature are often derived from the unaffected parts of mucosa that are resected during surgery for pathology in other parts of the intestine. It is obvious that these patients may have medical conditions that can act as confounders. In the present study, we collected enteroscopy jejunal biopsies from healthy normal-weight volunteers, who ingested an individualized well-controlled iso-caloric *high-carbohydrate diet* (HCD) and *high-fat diet* (HFD) in a randomized crossover fashion over two weeks each. An earlier report mainly concerned with systemic glycemic control was published elsewhere [5]. The present report deals with how these different, but isocaloric, diets influence SGLT1 expression in the human jejunal mucosa. We found that jejunal SGLT1 drastically increased during the high carbohydrate diet compared to during the fat-dominated one. This could be a variant of the long-term regulation of intestinal SGLT1, reviewed for example by Koepsell [2]. In another previous study from our laboratory, we showed that a high-fat diet in mice resulted in markedly raised jejunal expression of the ketogenesis rate-limiting enzyme mitochondrial 3-hydroxy-3-methylglutaryl-CoA synthase (HMGCS2), resulting in the production of ketone bodies like β-hydroxybutyrate (βHB) [6]. We therefore now wanted to investigate whether HFD, compared to HCD, could increase jejunal HMGCS2 expression also in normal-weight humans. Indeed, there was also a fat-induced HMGCS2-expression in the healthy jejunum. We then wanted to examine whether HFD-induced ketogenesis could have the potential to regulate SGLT1 expression. βHB is an effective class I histone deacetylase (HDAC) inhibitor [7], making it a mediator of epigenetic regulation through decreased histone deacetylation. Another signaling capacity of ketogenesis is its influence on the cell balance of the co-factor nicotinamide adenine dinucleotide (NAD) and thereby activation of the deacetylation enzymes sirtuins that are NAD-regulated class III HDACs [8,9]. Both these mechanisms have been reported to have effects on SGLT1 expression. Therefore, and based on the novel findings in the human biopsy material, we investigated the connection between intestinal ketogenesis and histone acetylation with SGLT1 expression in vitro in models of human jejunal enterocytes.

## 2. Methods

### 2.1. The Clinical Study

Details about the design of the study and other outcomes have been published previously [5] (see also CONSORT documents in Appendix A). Briefly, the sample size was calculated using the primary outcome (mucosal enlargement factor), but in the present report, one of the secondary outcomes is reported (SGLT1 transport in the Ussing chamber). The inclusion criteria were: voluntary participation, self-reported good general health status, age between 18 and 65 years, and body mass index (BMI) between 18 and 25 kg/m^2^. Exclusion criteria were: overweight or obesity (BMI > 25 kg/m^2^); history of drug abuse or smoking; use of prescription medications within the previous 14 days (with the exception of contraceptives); pregnancy or breastfeeding, or women at childbearing age not using adequate birth control. The study was designed as a single-centre, randomized, unblinded crossover study in healthy volunteers obtaining strictly controlled equicaloric diets either rich in carbohydrates (high-carbohydrate diet; HCD) or rich in fat (high-fat diet; HFD), with 60% energy from respective source. The diets were adjusted to the participant’s body weight according to the Mifflin St Jeor equation [10]. The order of the study diets was randomized in blocks due to cooking logistics. A study nurse randomly assigned the patients to start the study in one of the study diet blocks. The diets were administered over two 14-day periods in random order with an intervening wash-out period of a minimum of two weeks. The participants were instructed not to eat anything but the food and drinks provided by the laboratory but were free to drink extra tap water if needed. On day 12 of each diet period, the participants were subject to a mixed meal test, as reported elsewhere [5], and on day 14 of each diet period, the participants returned for enteroscopy after an overnight fast.

### 2.2. Enteroscopy

After sedation with midazolam and alfentanil, a thin-calibrated enteroscope was introduced into the gastroduodenum and proximal jejunum. Five to 10 biopsies were obtained from the jejunum approximately 50 cm distal to the ligament of Treitz. Four to six jejunal biopsies were either snap-frozen in liquid nitrogen or chemically fixated for later analyses (see Section 2.3). The remaining biopsies were prepared for functional assessments in mini-Ussing chambers. We only performed paired comparisons (HCD versus HFD), so when complete sets were not possible due to a lack of material or technical disturbances that individual had to be excluded from that particular analysis. The number of successful comparisons is given in the figures.

### 2.3. Immunofluorescence and Western Blotting

The biopsies were chemically fixed in phosphate-buffered 4% formaldehyde, dehydrated, embedded in paraffin, and cut in 5 μm sections. For a detailed description of immunofluorescence evaluation see Appendix A. The slides were analyzed using a fluorescence microscope (Leica Microsystems, Wetzlar, Germany). The frozen biopsies were used for western blot analyses and a detailed description is given in Appendix A. All antibodies mainly given appear in Appendix A. Data are presented as the ratio between the optical density of the primary antibody and GAPDH in each sample.

### 2.4. Ussing Chamber Experiments

Biopsies were then mounted in mini-Ussing chambers that had a biopsy insert with a diameter of 2 mm and an area of 0.034 cm^2^ (Warner Instruments, Hamden, CT, USA). The chamber was filled with 5 mL Krebs solution, bathing both the mucosal and serosal sides of the specimen. Short current pulses charge the epithelial capacitor, and when the current ends the capacitor is gradually discharged. The epithelial voltage response, as assessed from the discharge curve, and the magnitude of the applied current were used for the calculation of epithelial electrical resistance. Ohm’s law was used to get the epithelial net ion [11]. A detailed description of the Ussing chamber setup is given in Appendix A.

*Experimental protocol:* After an equilibration period of 10 min, basal values were recorded over 15 min. To be eligible for experimentation, each preparation had to exhibit a lumen-negative PD of at least 0.5 mV at baseline. To induce an electrogenic response, 10 mM D-glucose (Sigma-Aldrich, St. Louis, MO, USA) was added to the glucose-free solution in the luminal compartment. The net change in I_ep_ and R_ep_ was assessed after 5 min. To be included in the analysis at least two biopsies from each individual had to be successfully mounted, allowing at least one to serve as un-stimulated time-control to the glucose exposed preparations.

### 2.5. Caco-2 and Human Jejunal Enteroid Monolayer Cell Cultures

Caco-2 cells at passage 48 to 52 (Sigma-Aldrich) were seeded onto 12-well transwell membranes (3.0 μm pore size, Corning, Corning, NY, USA) and expanded in expansion media. The human jejunal enteroid culture was established from endoscopic biopsies taken from the jejunal portion of the small intestine from one healthy volunteer (ethics application number 049-16) using a technique developed by Sato et al. [12]. Differentiated monolayers of Caco-2 cells and small intestine-like human enteroids were cultured for 48 h in low concentration glucose (5.5 mM) media. To study the ketogenesis in the differentiated monolayers, different combinations of additives (butyrate, hymeglusin, nicotinamide, simvastatin) were given in low glucose (5.5 mM) media. After 48 h basolateral media was collected, βHB concentration in media was quantified according to β-Hydroxybutyrate Colorimetric Assay Kit (Cayman Chemicals, Ann Arbor, MI, USA) and proteins were extracted from cultures for western blot analysis. A detailed description is given in Appendix A.

### 2.6. Statistics

Statistical outcomes of western blot and Ussing chamber experiments with human biopsies were analysed with the Wilcoxon signed-rank test. The results from Caco-2 and human jejunal enteroid cultures were tested with the Mann-Whitney U test, or ordinary one-way ANOVA test with the Dunnett’s multiple comparisons test. A *p*-value of ≤0.05 was considered significant. All analyses were performed using Prism 9 for Mac OS X (GraphPad Software Inc., San Diego, CA, USA).

## 3. Results

Fifteen healthy volunteers (7 females, 8 males) with a mean bodyweight of 72 ± 3 kg ingested over 14 days the two iso-caloric diets; either a high-carbohydrate diet (HCD) or a high-fat diet (HFD) in a randomized order [5]. On day 14 in each dietary period, the volunteers were scheduled for enteroscopy with biopsy sampling from the jejunum.

### 3.1. Jejunal Epithelial Monosaccharide Transporters

Immunofluorescence analysis confirmed SGLT1 and GLUT5 to be predominantly expressed in the apical membrane of enterocytes, while GLUT2 expression was located more towards the basolateral side (Figure 1, upper panel). Immunofluorescence stainings for the individuals (with successful pairwise stainings) of the monosaccharide transporters SGLT1, GLUT2, and GLUT 5 are shown in Appendix A, Figure 1 (*n* = 8). HCD was associated with higher jejunal mucosal expression of the glucose-sodium transporter SGLT1 (*p* = 0.0003, *n* = 15) and the fructose transporter GLUT5 (*p* = 0.0004, *n* = 14) (Figure 1, lower panel). Glucose transporter GLUT2 showed higher expression following HFD (*p* = 0.0151, *n* = 15), whereas SGLT2, SGLT3 and GLUT1 were on similar levels (Figure 2, *n* = 15).

### 3.2. Glucose Induced a Mucosal Electrogenic Response after High-Carbohydrate Diet

Functional capacity test of SGLT1-mediated transport in human jejunal mucosa specimens using mini-Ussing chambers showed that baseline epithelial electrical resistance (R_ep_), as well as baseline epithelial current (Iep), were not different between the two diets (*n* = 11, Figure 2A, left and right panels, respectively). The addition of 10 mM D-glucose resulted in a similar effect on epithelial resistance for both diets (Figure 3B, left panel), whereas a higher (*p* = 0.0420, *n* = 11) response in epithelial current after HCD, compared to the HFD, was noted (*n* = 11, Figure 2B, right panel). This glucose-induced electrogenic response can be ascribed to SGLT1 as it is sensitive to phlorizin by more than 90% [13].

### 3.3. High-Fat Diet Increases HMGCS2 in the Jejunal Mucosa and Downregulates H3K9ac

Immunofluorescence of HMGCS2 revealed its expression was confined to the enterocytes of the villi (Figure 3A). All successfully paired stainings are shown in Appendix A (*n* = 7). Western blotting showed higher (*p* = 0.0006, *n* = 15) expression of the rate-limiting enzyme of ketogenesis, HMGCS2, in jejunal biopsies after HFD compared to HCD (Figure 3B, left panel). To examine a possible link between the decreased SGLT1 expression and increased expression of HMGCS2 induced by HFD, we checked the occurrence of acetylation of histone 3 at lysine 9 (H3K9ac). We found lower (*p* = 0.0245, *n* = 14) levels of H3K9ac following the HFD compared to HCD (Figure 3B, right panel). 

### 3.4. Jejunal GPR41 and GPR109A Are Upregulated following the High-Fat Diet

Western blot of free fatty acid and ketone body binding G protein-coupled receptors in the jejunal mucosa revealed that both GPR41 and GPR109A were significantly upregulated after HFD compared to HCD (*p* = 0.0002; *n* = 15 and *p* = 0.0353; *n* = 14, respectively; Figure 3C). The expression of the two purely free fatty acid-binding receptors, GPR40 and GPR43, was not different between the two diets (*n* = 14; Figure 3C).

### 3.5. Ketogenesis in Caco-2 and Human Jejunal Enteroid Cells

Caco-2 cells originate from the colon cancer mucosa of one patient but closely mimic small intestinal enterocytes both morphologically and functionally when cultured on semi-permeable membranes. These cells have been widely used as a model of the human small intestine. Human jejunal enteroid cultures are derived from stem cells in jejunal crypts, expanded as cysts in Matrigel (Figure 4A), and finally allowed to spontaneously differentiate into monolayers on semi-permeable membranes (Figure 4B). When the transepithelial electrical resistance after differentiation was above 1200 ohm/cm^2^, the membranes were considered to be fully confluent. The human jejunal enteroid monolayers at this stage display structural and functional characteristics of an epithelial membrane with brush-border microvilli and tight junctions (Figure 4C). 

During natural circumstances, the dietary fat (mostly triglycerides) will be luminally broken down to monoglycerides and free fatty acids by pancreatic lipases, and later “served” to the jejunal epithelium in the micellar form by the addition of biliary acids. In the present in vitro system, we simplified this system by exposing the epithelia to the short-chain free fatty acid *butyrate*, being the precursor for the formation of ketone bodies. In the basolateral media of Caco-2 cell cultures the concentration of βHB was substantial in presence of 10 mM butyrate (*n* = 21), but significantly lower (*p* ≤ 0.0001) after the addition of 1 μM of the HMGCS2 inhibitor hymeglusin (*n* = 21) (Figure 4D). In the human jejunal enteroid cultures, a similar dose of 1 µM hymeglusin resulted in higher variability in the results. However, if the dose was increased to 10 µM hymeglusin (*n* = 15) it significantly lowered (*p* ≤ 0.001) the concentration of βHB compared to 10 mM butyrate alone (*n* = 15), whereas the addition of 5 mM of sirtuin inhibitor nicotinamide (*n* = 12) had no effect (*p* = 0.8872, Figure 4E).

### 3.6. SGLT1 Expression and Acetylation of H3K9 Increase by Sirtuins Activation 

Expression levels of the SGLT1 protein were higher (*p* = 0.0403) in Caco-2 cells treated with 10 mM butyrate and 1 µM hymeglusin (*n* = 21) compared to butyrate only (*n* = 21). Moreover, treatment with 5 mM of the unspecific sirtuin inhibitor nicotinamide (*n* = 21) increased (*p* = 0.0002) SGLT1 expression in butyrate treated Caco-2 cells (Figure 5A). Acetylation of histone 3 at lysine (H3K9ac) measured by western blotting in the same cells showed a similar pattern to SGLT1. Levels of H3K9ac were significantly higher both after addition of hymeglusin and nicotinamide (*p* = 0.0122 and *p* = 0.0008, respectively), compared to controls with butyrate only (Figure 5B). The cholesterol pathway inhibitor simvastatin at concentration of 1 µM (*n* = 18) had no effect on either SGLT1, or on H3K9ac (*p* = 0.1157 and *p* = 0.9957, respectively, Figure 5A,B). The human jejunal enteroid cells displayed similar responses as the Caco-2 cells. Hymeglusin at a concentration of 10 µM (*n* = 9) increased the expression of both SGLT1 (*p* = 0.0089) and H3K9ac (*p* = 0.0008) (Figure 5C,D) compared to 10 mM butyrate alone (*n* = 11). Furthermore, following treatment with the sirtuin inhibitor nicotinamide at a concentration of 5 mM (*n* = 9) both SGLT1 and H3K9ac were expressed at higher levels (*p* = 0.0043 and *p* = 0.0037), compared to only butyrate (Figure 5C,D). Figure 5E displays a schematic representation of pathways that could explain these behaviors.

## 4. Discussion

In the healthy volunteers in the present study, the protein expression of jejunal monosaccharide transporters showed a well-known diet-dependent pattern; SGLT1 and GLUT5 decreased during the high-fat diet compared to the high-carbohydrate diet. The functional Ussing chamber experiments underscore these results; there was a decreased glucose-induced epithelial current after the high-fat compared to the high-carbohydrate diet. This glucose-induced current was SGLT1-dependent due to its sensitivity to phlorizin, a SGLT1-specific inhibitor [13,14]. Both SGLT1 and GLUT5 are located in the apical membrane of the enterocyte, which was confirmed by immunofluorescence. SGLT1 transports glucose and galactose with sodium and GLUT5 transports fructose into the enterocyte [1,15]. The expression of GLUT2 was basolateral and its function is classically to deliver glucose from the enterocyte to the enteric venous and eventually to the portal blood [2]. GLUT2 has been shown to translocate to the apex of enterocytes under conditions with a high luminal concentration of glucose [16,17,18], but conflicting data has also been presented showing that GLUT2 is maintained in its basolateral position [1,19]. In the current study, GLUT2 was observed to be more prominent in the basolateral compartment of the enterocytes. However, the biopsies were taken after 10–12 h of fasting and any translocation of GLUT2 to the apical membrane was not to be expected. Thus, the present data do not directly give support to either of these divergent models. We found GLUT2 expression to be increased in the high-fat situation, but still with a basolateral location as detected by immunofluorescence. During a high-fat diet, there is a relative shortage of rapid carbohydrates in the luminal content of the intestine and GLUT2 might be increased basolaterally in order to allow glucose transport from the blood to the cells. This is because the epithelium itself requires glucose for its internal metabolism, which was shown e.g., after bariatric surgery [20,21]. When taken together, the Ussing chamber experiments, the western blotting, and the immunofluorescence show a diet-dependent adaptation of the human jejunum. The absorption of glucose increases via SGLT1 after the high-carbohydrate diet but becomes down-regulated by the high-fat diet (i.e., the luminal energy has a different composition). Factors initiating downregulation of SGLT1 during situations with low glucose availability have been discussed over a long period of time. Epigenetic modifications, as well as post-translational regulation influenced by insulin, glucagon, incretins, angiotensins, and other factors, have been suggested [3,22,23]. We now propose intestinal ketogenesis as a new factor that can decrease SGLT1 expression and glucose absorption in the human small intestine.

Ketone bodies are known to appear in the human body when not enough energy can be absorbed from the gut. During starvation, the hypoglycemia and the following lipemic response increase HMGCS2 expression in the liver in order to support the brain with fuel in the form of ketone bodies. A very high systemic concentration of ketone bodies (keto-acidosis) is a rare consequence of untreated diabetes. The condition is dangerous because the concentration of ketone bodies can reach lethal levels. Intestinal ketogenesis is much less explored, but reports appear in the literature during the last decades. For example, HMGCS2 is expressed in the colon and metabolizes butyrate produced by the luminal microbiota [24]. In rats, HMGCS2 is expressed and ketone bodies are produced in the small intestine during the postnatal-preweaning period but decrease after weaning [25]. Interestingly, in the adult rat small intestine, the expression of HMGCS2 can re-occur following a high-fat diet [26]. 

We showed in a recent paper that HMGCS2 and its metabolite βHB were increased specifically in the jejunal mucosa following a fatty diet in mice and that both decreased substantially after Roux-en-Y gastric bypass surgery in humans [6]. In the present study, we found a significant increase in the expression level of HMGCS2 in the human jejunal mucosa after the high-fat diet compared to after a high-carbohydrate diet. We were not able to measure βHB in these volunteers (due to a shortage of material), but aside from the increased expression of HMGCS2 we also found increased expression of the ketone body/fatty acid receptors GPR41 and GPR109A [27]. Furthermore, the jejunal HMGCS2 was located in the middle parts and tips of the villi, thus having the same location as SGLT1 in the same area where lumen-degraded carbohydrates are absorbed. Therefore, we hypothesized that the increased mucosal ketone bodies could be responsible also for the decreased SGLT1. The acetylation of histone 3 at lysine 9 (H3K9ac) has been shown to induce increased expression of the SGLT1 gene in mice fed a high carbohydrate-to-fat-ratio diet [28,29,30]. In the present study in man, we found increased acetylation of histones, measured as H3K9ac, after the high-carbohydrate diet compared to after the high-fat diet. Ketone bodies have been reported to exert a histone deacetylase (HDAC)-inhibiting effect and would thus increase H3K9 acetylation and SGLT1 expression [7]. We suggest, therefore, that the acetylation effect is not via a generalized ketone body mediated HDAC-inhibition. Ketogenesis (the synthesis of ketone bodies) may alter the balance between the oxidized and reduced forms of NAD, NAD^+,^ and NADH, respectively. The result of increased ketogenesis should be a shift towards NAD^+^ because it is a product of the enzymatic conversion of the ketone body acetoacetate to βHB. An increased amount of NAD^+^ is known to control the activity of sirtuins, which belong to class III HDACs and that have the capacity to deacetylase H3K9 [8,9]. Histone- and protein-acetylation have a great impact on metabolism and are related to many diseases, such as type 2 diabetes mellitus [31]. Therefore, ketogenesis may possess opposing signaling effects on acetylation enzymes: either inhibition (of class I HDACs) via βHB or activation (of sirtuins) through NAD^+^. To be able to investigate these mechanistic possibilities, we changed to in vitro models. Since the acetylation of histone increased with HFD in the biopsy material we suspected the mechanism to be regulated through sirtuins rather than class I HDACs.

We used monolayers of Caco-2 cells and human jejunal enteroids to investigate the link between ketogenic activation of sirtuins and a histone acetylation effect on SGLT1 expression. In order to mimic the high fat-diet condition, we cultured the model systems in presence of the short-chain fatty acid butyrate. This metabolite is close to the end of β-oxidation and will result in rapid production of acetyl-CoA. Butyrate is also processed by fermentation of indigestible fibres by colonic bacteria [32]. The addition of butyrate to the in vitro model resulted in a substantial formation of the ketone body βHB, which we then were able to inhibit using the HMGCS2-inhibitor hymeglusin [33]. The regulation of SGLT1 expression was tested with the addition of butyrate only as the control. The addition of hymeglusin or nicotinamide [34] increased SGLT1 expression and acetylation of H3K9 in both the Caco-2 and the enteroid cultures. This suggests sirtuin-mediated deacetylation of the histone in the fatty diet situation resulting in the decreased SGLT1 expression. This type of sirtuin awaits further study. Involvement of the jejunal microbiota seems unlikely. First, very few microbials appear to inhabit this segment of the small intestine [35]. Secondly, the in vitro experiments were performed without any microbial components. Butyrate is, as mentioned, a microbial metabolite in the colon but was given as a substrate for the production of acetyl-CoA. Taken together, we believe that the acetylation effect in the villous enterocytes is not dependent on the luminal microbiota. The doses of hymeglusin [33], nicotinamide [34], and simvastatin [36] used were taken from the literature and might not be optimal for this setting. Still, the effects exerted by these drugs bring more insight into the potential fatty acid-induced downregulation of SGLT1. The HMGCoA-reductase inhibitor simvastatin did not affect SGLT1 expression or acetylation of H3K9 in Caco-2 cells, excluding potential effects of hymeglusin being mediated through the mevalonate pathway. The results contrast somewhat with the intestinal ketogenesis studies on cell differentiation performed in Caco-2 cells and mice made by Wang et al. [37] and the stem cell homeostasis in mice by Cheng et al. [38]. These studies were concentrated around aspects of βHB contributing to intestinal cell differentiation and adaption to diet. Here we suggest effects over the NAD^+^/sirtuin axis that is acting in parallel with the ketone bodies. Furthermore, we studied the villi in healthy human jejunum, mainly the enterocytes, in a region where the monosaccharide absorbers are located. Because the enterocytes move rapidly from the crypt region up towards the tips of the villi, it would be interesting to see if they are premade in the crypt region, or if they differentiate along the villous axis. 

## 5. Conclusions

In healthy, normal-weight volunteers we show that the glucose transporter SGLT1 is reduced, whereas the ketogenic enzyme HMGCS2 is increased in the jejunal mucosa by a high-fat diet compared to an iso-caloric high-carbohydrate diet. This was further corroborated by functional assessments using mini-Ussing chambers. Furthermore, we found lower acetylation of histone 3 at lysine 9 (H3K9ac) after the fatty diet, suggesting a decreased SGLT1 gene transcription. In vitro, in Caco-2 cells and human jejunal enteroid cultures, a fatty acid increased ketogenesis. However, rather than a direct effect of ketone bodies themselves, it was found that the simultaneously increased tissue levels of NAD^+^, in turn, activated the *sirtuins* to induce a de-acetylation of histone, thereby reducing the expression of SGLT1. 

## Figures and Tables

**Figure 1 nutrients-14-01961-f001:**
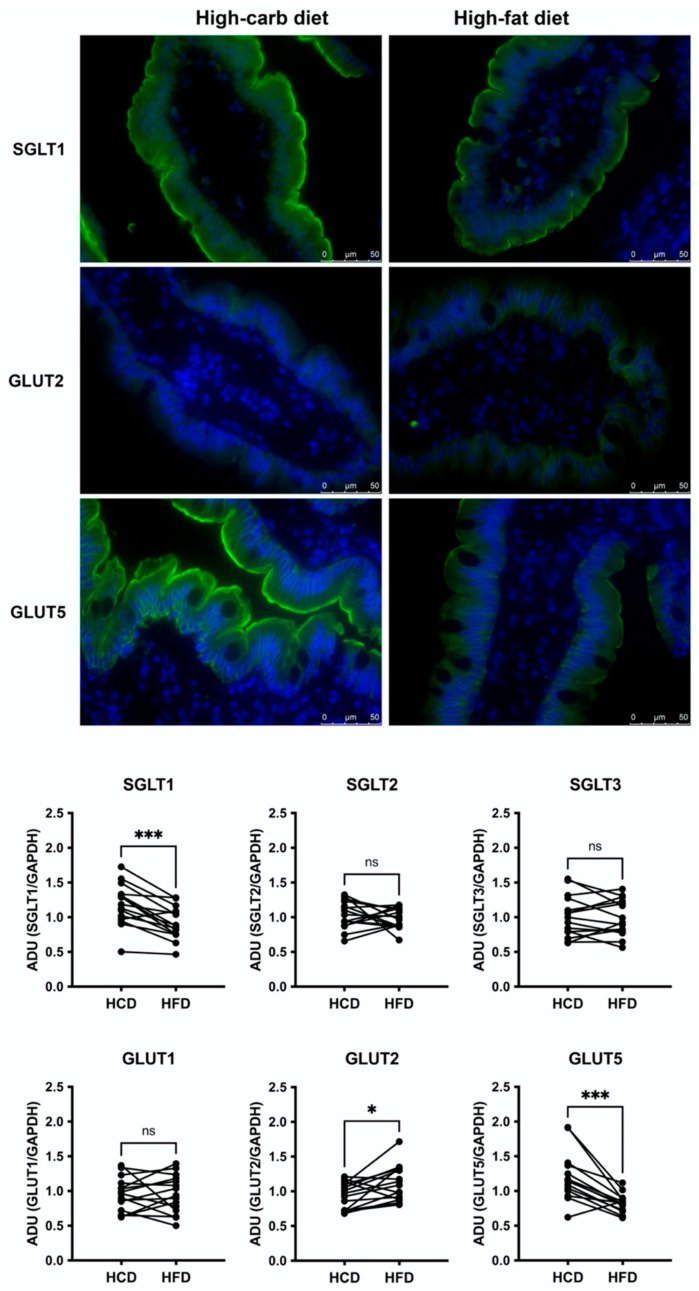
Upper panel: Immunofluorescence of SGLT1, GLUT2, GLUT5 in jejunal biopsies after 2 weeks of high-carbohydrate (left) or high-fat diet (right). Each protein is represented with staining from one subject. Lower panel: Western blot analyses of monosaccharide transporters SGLT1, SGLT2, SGLT3, GLUT1, GLUT2, and GLUT5 in human jejunal mucosal biopsies obtained after two weeks of high-carbohydrate (HCD) or high-fat diets (HFD). Data are related to a housekeeping protein (GAPDH) and pair samples are reported. *n* = 15 for all except for GLUT5 where *n* = 14. *p*-values were calculated with Wilcoxon signed-rank test and reported for each protein. * *p* ≤ 0.05, *** *p* ≤ 0.001, ns: non-significant difference.

**Figure 2 nutrients-14-01961-f002:**
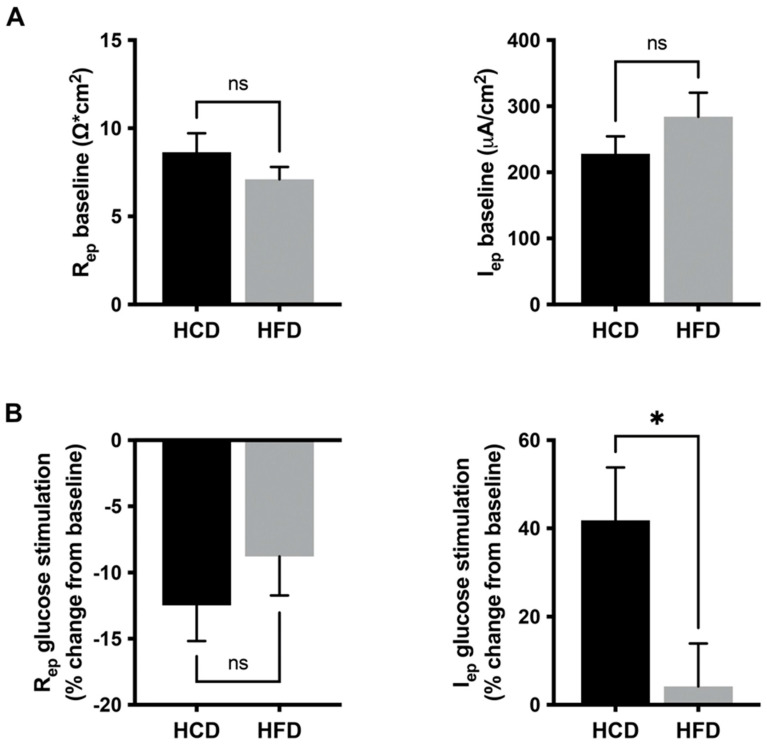
Jejunal mucosal specimens mounted in Ussing chambers after carbohydrate or fat-dominated diets. In (**A**), baseline epithelial electrical resistance (R_ep_) and epithelial current (I_ep_) are shown, respectively. In (**B**), the responses to 10 mM D-glucose at the luminal side are given. Note that the epithelial current (I_ep_) response to 10 mM glucose is significantly larger after the high-carbohydrate diet (HCD) compared to the high-fat diet (HFD). Data are given as mean + SEM; *n* = 11 successful pairs of mucosae; Wilcoxon signed-rank test. * *p* ≤ 0.05, ns: non-significant difference.

**Figure 3 nutrients-14-01961-f003:**
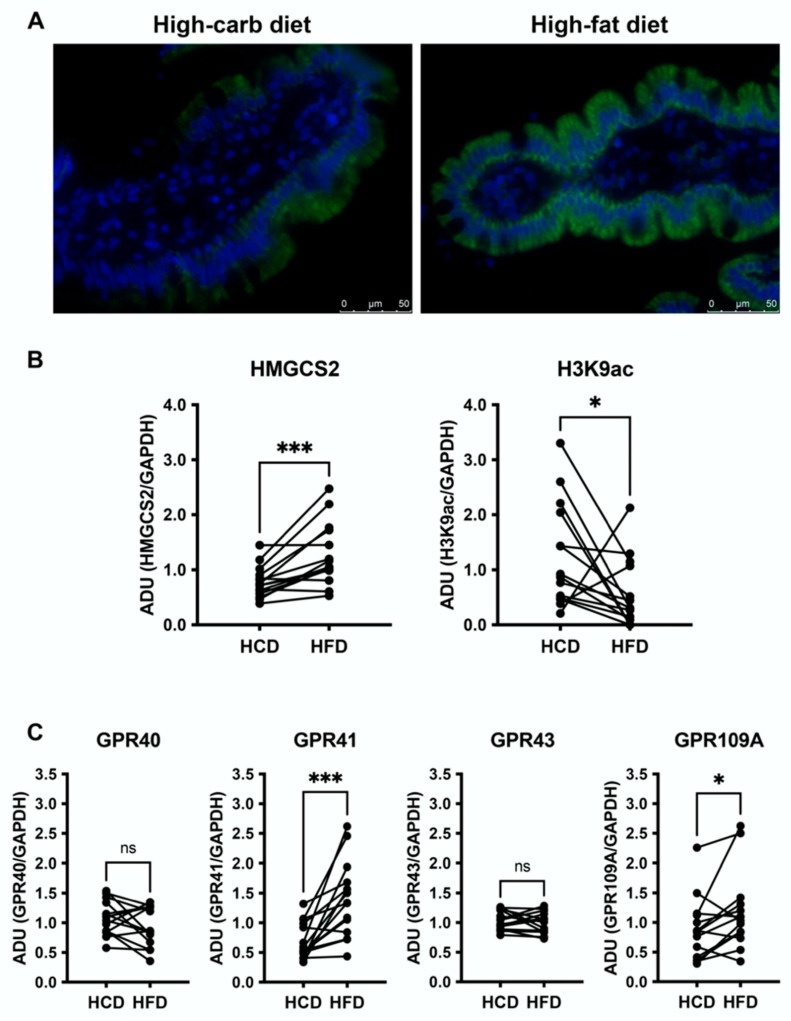
(**A**): Immunofluorescence of the ketogenesis rate-limiting enzyme HMGCS2 in the jejunal mucosa after two weeks of high-carbohydrate (HCD, left) and high-fat diet (HFD, right). Paired sample. (**B**): Western blot analyses of HMGCS2 and acetylation of histone 3 at lysine 9 (H3K9ac) in human jejunal mucosal biopsies obtained after HCD or HFD. (**C**): Western blot analyses of free fatty acid receptors GRP40, GPR41, GPR43, GRP109A in human jejunal mucosal biopsies obtained after two weeks of HCD or HFD. Data are related to a housekeeping protein (GAPDH) and paired samples are reported. *p*-values were calculated using Wilcoxon signed-rank test, *n* = 15 for HMGCS2, GPR41, and *n* = 14 for H3K9ac, GRP40, GRP43 and GRP109a. * *p* ≤ 0.05, *** *p* ≤ 0.001, ns: non-significant difference.

**Figure 4 nutrients-14-01961-f004:**
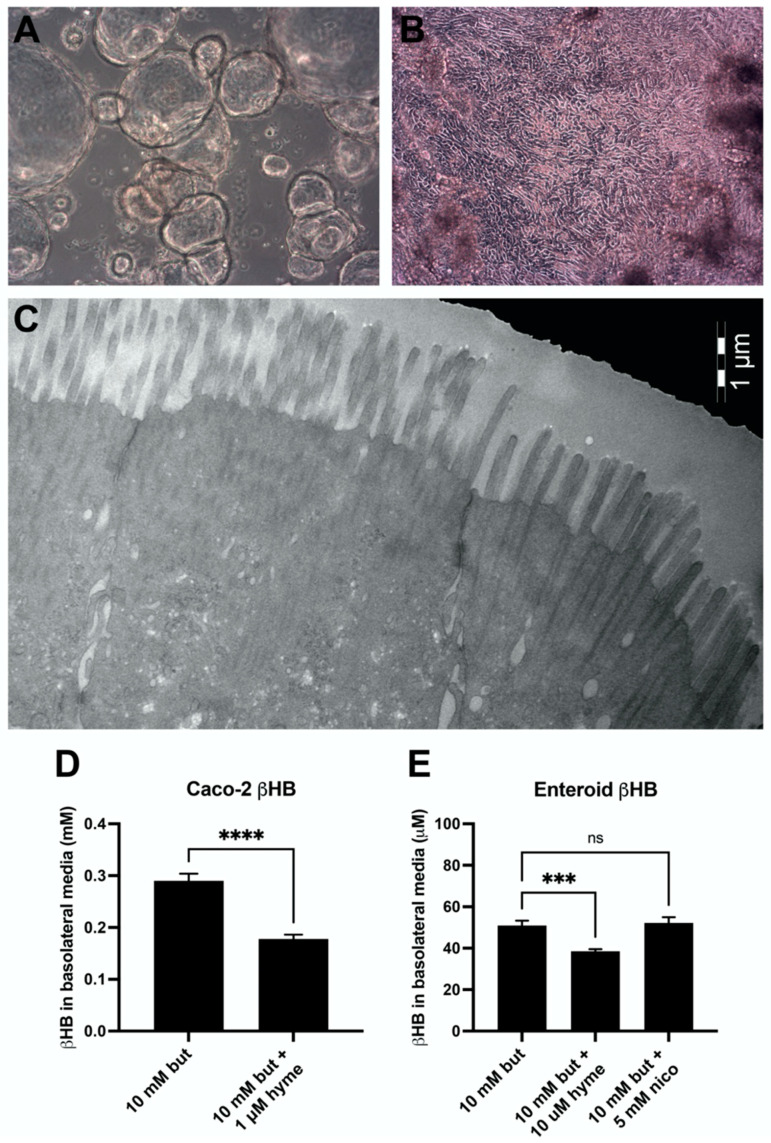
Phase contrast micrographs of enteroids in cystic (**A**) and membrane (**B**) culture. In (**C**), transmission electron microscopy image of a human enteroid differentiated on membranes showing microvilli structures and tight junctions is shown. Quantification of βHB in basolateral media after 48-h stimulation with butyrate (*n* = 21 and 15, for respective model), butyrate and hymeglusin (*n* = 21 and 15), and butyrate and nicotinamide (*n* = 12, only for enteroids) of differentiated monolayers of Caco-2 cells (**D**) and human jejunal enteroids (**E**). Data are shown as mean + SEM; Mann-Whitney test. *** *p* ≤ 0.001, **** *p* ≤ 0.0001, ns: non-significant difference.

**Figure 5 nutrients-14-01961-f005:**
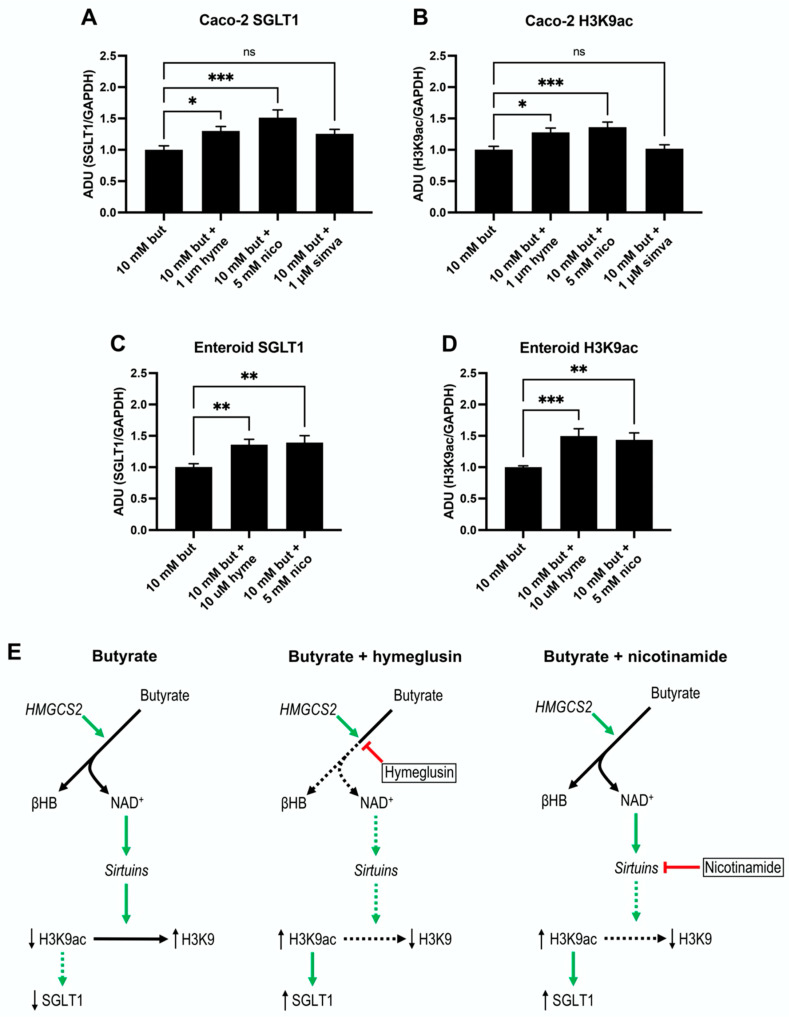
Western blot analysis of SGLT1 (**A**,**C**) and acetylated histone 3 at lysine 9 (H3K9ac, (**B**,**D**)) in differentiated monolayers of Caco-2 and jejunal enteroids after 48-h culture in butyrate (*n* = 21 for Caco-2 and *n* = 11, for enteroids), butyrate and hymeglusin (*n* = 21 for Caco-2 and *n* = 9 for enteroids), butyrate and nicotinamide (*n* = 21 Caco-2 and *n* = 9 for enteroids) or butyrate and simvastatin (*n* = 18, only Caco-2). All data are given as mean + SEM. Western blot data are related to a housekeeping protein (GAPDH). *p*-values were calculated using ordinary one-way ANOVA test with Dunnett’s multiple comparisons test against groups treated with only butyrate. * *p* ≤ 0.05, ** *p* ≤ 0.01, *** *p* ≤ 0.001, ns: non-significant difference. In (**E**), a proposed mechanistic regulation of H3K9ac and SGLT1 for each group is presented. Black, green and red arrows indicate reactions, stimulation, and inhibitions, respectively. Italics indicate enzymes. Dashed lines indicate decreased production, decreased activation, or decreased inhibition.

## Data Availability

All authors had access to the study data and reviewed and approved the final manuscript. All data relevant to the study are included in the article or uploaded as Appendix A. Individual participant data will not be shared.

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
