# Peer review of "A Fatty Diet Induces a Jejunal Ketogenesis Which Inhibits Local SGLT1-Based Glucose Transport via an Acetylation Mechanism—Results from a Randomized Cross-Over Study between Iso-Caloric High-Fat versus High-Carbohydrate Diets in Healthy Volunteers"

_nutrients, 2022, doi:10.3390/nu14091961_

Round 1

Reviewer 1 Report

Introduction to the topic is very long, given that the earlier clinical trial is already published. authors need to cut down severely to stick to the main point of the research.

Pl explain what does authors refers by “Chemically fixated”.

Abstract is misleading, this reviewer thought that they ran a clinical trial for this manuscript. Please clearly mention that this manuscript is secondary analysis of a previously published clinical trial samples.

The cell culture data from Caco cell-line adds nothing to this paper nor to the discussion, as the same experiments give similar results and conclusion with enteroid culture experiments.

Figure 6 is redundant as similar information is already presented in Figure 5 E.

Author Response

We thank the Reviewer for his encouraging work!

  1. We have now removed most of the previous report (ref 5) and the Introduction  is now shortened by at least 20%.  
  2. The "chemically fixated" (2.2) is now referred to the next paragraph (2.3) where it is described in more detail.
  3. Thank you! We agree to that the Abstract is misleading. We have now corrected this by putting in some extra words in the Abstract. 
  4. We understand the reviewer, but we have decided not to delete the Caco-cells. The main reason is that the simvastin-experiments were done in these cells, thereby excluding the "cholesterol pathway".
  5. We have taken out Figure 6, its legend and one sentence in Discussion. 

Reviewer 2 Report

Manuscript Number: nutrients-1702354

Title: A fatty diet induces a jejunal ketogenesis which inhibits local

SGLT1-based glucose transport via an acetylation mechanism –

results from a randomized cross-over study between iso-caloric

high-fat versus high-carbohydrate diets in healthy volunteers

Congratulation to this interesting article.

It is very nice structured, showing a human study and then mechanistical insights in vitro.

The proposed mechanism seems is very likely.

Minor comments:

  1. You mention your previous study on patients with Roux-en-Y gastric bypass surgery. Maybe you can check this, there might be action of ghrelin that is secreted I the stomach on GLP-1. That might be added to the meachnism as well. (e.g. Zhang et al. 2020)
  2. Please add also the dangers of a metabolic acidosis that can accur with too high amounts of keton bodies
  3. Please add also the important role of SGLT-1 in water homeostasis that is important to be considered as well.

Author Response

We thank the Reviewer for the nice words. 

  1.  Reviewer-1 wanted a better focus on the rationale of the present study, and a reduction of the length of the Introduction. Therefore, our previous study with Roux-en-Y gastric by pass patients was deleted, with exception the findings of fat given to mice (Introduction) and the ketone bodies after RYGP (Discussion). We hope for understanding. 
  2. Thank you! We have given some sentences on the danger of keto-acidosis in the Discussion.
  3. We have added this important SGLT1-variable in Introduction. However, we have not written anything on its role in oral-rehydration. This was done in order not to diffuse the subject of the manuscript (monosaccarides).